



# Effects of Aerosol Size and Coating Thickness on the Molecular Detection using Extractive Electrospray Ionization

*Chuan Ping Lee,[1] Mihnea Surdu,[1] David M. Bell,[1] Houssni Lamkaddam,[1] Mingyi Wang,[2,3] Farnoush Ataei,[6] Victoria Hofbauer,[2,3] Brandon Lopez,[2,4] Neil M. Donahue,[2,3,4,5] Josef Dommen,[1] Andre S. H.*
*Prevot,[1] Jay G. Slowik,[1] Dongyu Wang,[1,\*] Urs Baltensperger,[1] Imad El Haddad[1,\*]*

[1]Laboratory of Atmospheric Chemistry, Paul Scherrer Institute (PSI), 5232 Villigen, Switzerland.
[2]Center for Atmospheric Particle Studies, Carnegie Mellon University, Pittsburgh, PA, 15213, USA.
[3]Department of Chemistry, Carnegie Mellon University, Pittsburgh, PA, 15213, USA.
[4]Department of Chemical Engineering, Carnegie Mellon University, Pittsburgh, PA, 15213, USA.
[5]Department of Engineering and Public Policy, Carnegie Mellon University, Pittsburgh, PA, 15213, USA
[6]Leibniz-Institute for Tropospheric Research, 04318 Leipzig, Germany.

Correspondence to: Imad El Haddad (imad.el-haddad@psi.ch), Dongyu Wang (dongyu.wang@psi.ch), Jay G. Slowik (jay.slowik@psi.ch)

**Abstract.** Extractive electrospray ionization (EESI) is a well-known technique for high throughput online molecular characterization of chemical reaction products and intermediates, detection of native biomolecules, in vivo metabolomics, and environmental monitoring with negligible thermal and ionization-induced fragmentation for over two decades. However, the EESI extraction mechanism remains uncertain. Prior studies disagree whether analyte particles between 20 and 400 nm diameter are fully extracted or if the extraction is limited to the surface layer. Here, we examined the analyte extraction

mechanism by assessing the influence of analyte particle size and coating thickness on the detection of the molecules therein. We find that analyte particles are extracted fully: Organics-coated $NH_4NO_3$ particles with a fixed core volume (156 and 226 nm in diameter without coating) show constant signals for $NH_4NO_3$ independent of the shell coating thickness, while the signals of the secondary organic molecules comprising the shell varied proportionally to the shell volume. We also find that the EESI sensitivity exhibits a strong size dependence, with an increase in sensitivity by one to three orders of magnitude as

analyte particle size decreases from 300 nm to 30 nm. This dependence varies with the electrospray (ES) droplet size and the analyte particle residence time in the EESI inlet, suggesting that the EESI sensitivity is influenced by the coagulation rates between analyte particles and ES droplets. Overall, our results indicate that, in the EESI, analyte particles are fully extracted by the ES droplets regardless of the chemical composition, when they are collected by the ES droplets. However, their coalescence is not complete and depends strongly on their size. This size-dependence is especially relevant when EESI is used

to probe size-varying analyte particles as is the case in aerosol formation and growth studies with size ranges below 100 nm, while it does not significantly influence the detection of ambient aerosol dominated by particle sizes ranging between 100 - 2500 nm, i.e. the accumulation mode.

## 1 Introduction

Atmospheric aerosols are suspended particles in the air ranging from few nanometers (nm) to several micrometers (μm) in
diameter. Fine particles (< 1 μm) comprise nucleation, Aitken and accumulation mode particles, and account for 50-70 % of the total particulate matter (PM) mass concentrations in polluted environments (Yue et al., 2009). They can affect the earth's radiative balance either directly, by interacting with solar radiation, or indirectly by acting as cloud condensation nuclei (CCN), influencing cloud albedo and lifetime (Steinfeld 1998). Exposure to PM is one of the leading causes for premature death, accounting for ~8.9 million deaths, or ~10% of total global burden of mortality in 2015 (Burnett et al., 2018), though the
underlying mechanisms remain uncertain (Daellenbach et al., 2020). PM can be emitted as primary aerosol or produced in the atmosphere after chemical reactions via nucleation or condensation of gas-phase products (Clarke et al., 1984; Hoffmann et



al., 1997; Kalberer et al., 2004; Berndt et al., 2005; Jimenez et al., 2009; Kirkby et al., 2011). Heterogeneous reactions may also further increase the complexity of ambient aerosol mixtures (George and Abbatt 2010; Ditto et al., 2020).

Online molecular characterization of atmospheric aerosols is required to resolve the spatiotemporal variability of PM molecular

composition and to identify PM sources. Progress has been made with the development of chemical ionization interfaces such as the Filter Inlet for Gases and AEROsols (FIGAERO) (Lopez-Hilfiker et al., 2014), Thermal Desorption Differential Mobility Analyzer (TD-DMA) (Holzinger et al., 2010; Wagner et al., 2018), and Chemical Analysis of Aerosol Online (CHARON) (Eichler et al., 2015) coupled to a mass spectrometer. However, these techniques suffer from thermal decomposition of the analyte prior to ionization and/or ionization-induced fragmentation, impeding molecular speciation

(Müller et al., 2017; Stark et al., 2017). To complement these instruments, an extractive electrospray (ES) ionization time-of-flight mass spectrometer (EESI-TOF) was developed to enable molecular characterization of organic aerosol at 1 Hz time resolution with ng m$^{-3}$ level detection limit, and minimal thermal and ionization-induced fragmentation (Lopez-Hilfiker et al., 2019). The EESI-TOF was further developed to enable online tandem mass spectrometry for molecular structure elucidation, and to characterize water-soluble metals (Giannoukos et al., 2020; Lee et al., 2020).

Several studies such as extraction of macromolecules from colloidal solution (Chen et al., 2006), electron-transfer-catalyzed dimerization (Marquez et al., 2008), and gas plume mixing in the charged droplets (Cheng et al., 2008), reported that the ionization of EESI mainly happens in the liquid phase via interaction between ES charged droplets and neutral analyte droplets. Due to the lower water content (< 50%) of our analyte droplet in all our experiments than a typically known water content of droplet (> 90 %), we refer our analyte droplets as analyte particles here hereafter. If this liquid-phase extraction of EESI occurs

via total coalescence between analyte particles and ES droplets, the measured EESI signal should be proportional to the total analyte mass concentration, i.e. full extraction of analyte particles by ES droplets as demonstrated by several works (Law et al., 2010; Lopez-Hilfiker et al., 2019). In contrast, new studies suggested that the analyte particles are only partially probed, limiting the full quantification of the extracted analyte with electrospray ionization (Wang et al., 2012, Kumbhani et al., 2018). Kumbhani et al. suggested that only the surface of particles with a diameter size of approximately 100 nm is extracted by

comparing infusion ESI-MS with EESI-MS using coated chemical standards (Kumbhani et al., 2018). Using other techniques such as phase Doppler anemometer, Wang et al. suggested that the extraction happens via fragmentation of the analyte and ES droplets (Wang et al., 2012). Finally, other studies have proposed that the EESI extraction efficiency could depend on the analyte volatility and size (Meier et al., 2011a; Pagonis et al., 2020). Since all of these studies only probed simple systems i.e. individual chemical standards using one kind of experimental and EESI ionization source, these discrepancies could be

inherently attributed to their differences of ES ionization geometries, experimental conditions, irreproducible ES Taylor cone conditions and perhaps the choices of the chemicals.

Nevertheless, without reconciling the discrepancies of these reported EESI mechanisms, the EESI quantification must be regarded as highly uncertain when the technique is used to probe varying size distributions of analyte particles that exist in different mixing states and are comprised of different molecular polarity, volatility, and sizes. Here, we take advantage of

recent advancements in analyte particle generation and chemical analysis to evaluate the extraction mechanism of EESI using three different methods for analyte particle generation and several online mass spectrometers for aerosol online chemical analysis. First, we characterize the EESI extraction efficiency with analyte particles containing atmospherically relevant standard compounds and mixtures, selected in the size-range 30-500 nm using an aerosol aerodynamic classifier. We elucidate the influence of the ES operating parameters and residence time between charged and analyte particles using two different

EESI sources. Second, we assess whether the EESI extraction efficiency depends on the analyte chemical composition, by comparing EESI-TOF and a chemical ionization (CI) TOF-MS equipped with a Filter Inlet for Gases and AEROsols (FIGAERO) sampling manifold (FIGAERO-CI-ToF-MS) measurements for α-pinene secondary organic aerosol (SOA) generated in the CLOUD (Cosmics Leaving OUtdoor Droplets) chamber at CERN, Switzerland (Kirkby et al., 2016; Tröstl et al., 2016; Dias et al., 2017). Third, we determine whether analyte particles are fully extracted or if extraction is limited to the



coated surface by coating monodisperse $NH_4NO_3$ particles at a fixed size with variable amounts of oxidation products in an oxidation flow tube reactor.

## 2 Experimental

### 2.1 Materials

Acetonitrile (Sigma-Aldrich, UV grade), sodium iodide (Sigma-Aldrich, 99.7% purity) and milli-Q water (18 MΩ cm) were
used to prepare the electrospray (ES) and chemical standard nebulization solution. Polyimide-coated fused silica capillary (i.d. 75 μm, o.d. 369 μm; BGB Analytik, Boeckten, Switzerland), HEPA capsule filter (Pall Corporation), PEEK tubing (i.d. 500 μm, o.d. 1/16 inch; BGB Analytik, Boeckten, Switzerland) and charcoal denuders (Ionicon GmbH, Austria) were used for the electrospray ionization inlet. As chemicals, α-Pinene (Sigma-Aldrich, 99% purity), levoglucosan (Sigma-Aldrich, 99% purity), sucrose (Sigma-Aldrich, 99% purity), ammonium nitrate (Sigma-Aldrich, 98% purity) were used.

### 2.2 Electrospray ionization configuration

Two designs of the EESI sources with a factor of 2 difference in their residence time in the electrospray ionization region were used in this experiment, in front of a high-resolution TOF mass spectrometer (HTOF, Tofwerk AG, Switzerland). EESI source A (Lopez-Hilfiker et al., 2019) and B were developed for Tofwerk TOF and Thermo Scientific Orbitrap mass analyzers (Figure S1), respectively, though EESI source B is compatible for both mass analyzers, as described in details elsewhere (Lee et al.,
2020). Source A was used throughout the whole experiment and Source B was only used in the study of Figure 2. Two electrospray (ES) solutions were used to generate charged ES droplets: (1) acetonitrile/$H_2O$ (50/50 v/v); and (2) 100% $H_2O$ (Table S1). Both solutions were doped with 100 ppm NaI. A potential difference of around 2.6-2.9 kV relative to the MS interface was applied to the ES solution, and an air pressure difference of 120 to 800 mbar was applied to the ES solution bottle reservoir, delivering 0.3 - 23 μl min$^{-1}$ of ES solution via a polyimide fused silica capillary (OD: 369 μm and ID: 50, 75
and 100 μm, BGB Analytik, Switzerland). Different ES operating parameters with estimated ES parent droplet size ranging from 0.7 – 5.66 um are tabulated in Table S2. The ES droplets intersected with the analyte particles before entering the heated TOF-capillary kept at 275 ˚C (<1 ms residence time), undergoing a Coulomb explosion as the ES droplets evaporated. The generated ions from organic molecules are detected by the HTOF predominantly (> 95 % relative abundance) as sodiated adducts ($[M+Na]^+$) in the positive ionization mode. Ammonium nitrate ($NH_4NO_3$), an inorganic salt, is detected as
$[NaNO_3+Na]^+$. The raw mass spectra (1 Hz) were post-averaged every 10 seconds using Tofware (version 2.5.13). All measured analyte signals were normalized by the most abundant electrospray ion (i.e. $[NaI+Na]^+$) to account for the variation of electrospray signal (± 5 %).

### 2.3 Analyte particle size selection

Figures S2 and S3 show two experimental setups for the investigation of size-dependent of analyte particle extraction
efficiency using EESI. Chemical standards were used in the first experimental setup (Figure S2). Three individual aqueous solutions containing 4000 ppm of levoglucosan, sucrose and ammonium nitrate, respectively, were nebulized separately at 1.4 L min$^{-1}$, which was then mixed with a 1.6 L min$^{-1}$ make-up zero air. The output analyte particles were dried with a custom-made drier containing silica gel and subsequently size selected using an aerosol aerodynamic classifier (AAC; Cambustion, United Kingdom) to produce monodisperse analyte particles (Tavakoli and Olfert 2013, 2014; Tavakoli et al., 2014). The size
selection was implemented by centrifugal separation of the analyte particles according to their mass. Unlike size selection using differential mobility analyzers (Lopez-Hilfiker et al., 2019), size selection using the AAC does not require electrical charging, thereby avoiding multi-charging artifacts. The possible multi-charging on analyte particles might affect the normal extraction condition by the EESI where the analyte particles are assumed to be neutral (Kebarle and Verkcerk 2009, 2012). In





addition, doubly charged analyte particles could result in underestimation of analyte particle size and mass concentration.

Therefore, the new experimental setup we propose here is very well suited for the study of the EESI size-dependence sensitivity.

After analyte particle size selection, the sample was drawn through a multichannel charcoal denuder to strip gas-phase constituents before entering the EESI-TOF inlet manifold. The sample was also characterized immediately upstream of the electrospray region by a nano-scanning mobility particle sizer (size range 2.5 - 239 nm, nano-SMPS, TSI Inc., USA), a

scanning mobility particle sizer (size range 16 - 638 nm, TSI SMPS, TSI Inc., USA) and an aerosol mass spectrometer equipped with a long time-of-flight mass analyzer (AMS-LTOF, Aerodyne Research Inc., USA) (Figure S2). The high concentration of the chemical solutions ensured that sufficient analyte concentrations (> 3 $\mu$g m$^{-3}$) remain after size-selection by the AAC using the highest possible sheath flow (15 L min$^{-1}$ at an aerodynamic diameter $D_{ae}$ > 150 nm) to produce highly monodisperse analyte particle distributions (Tavakoli and Olfert 2014). The resulting electrical mobility diameters after size selection by the AAC

ranged from 34 to 500 nm (Figure S5).

In the second configuration (Figure S3), we investigated the size-dependent sensitivity of the EESI using biogenic SOA produced from $\alpha$-pinene oxidation in the Cosmic Leaving OUtdoor Droplets (CLOUD) chamber at CERN, Switzerland ( Kirkby et al., 2011, Dias et al., 2017), at -50 to -30 °C (Simon et al., 2020). The EESI-TOF signals of individual $\alpha$-pinene oxidation products (C$_{10}$H$_{16}$O$_{3-8}$) were compared to FIGAERO-CI-ToF-MS (Lopez-Hilfiker et al., 2014). The FIGAERO-CI-

ToF-MS measured both the gas- and particle-phase. Here, particles are first collected onto a 24 mm ø PTFE filter via a dedicated port with a sampling flow rate of 6 L min$^{-1}$. Then, 2.7 L min$^{-1}$ of pure N$_2$ is heated progressively to thermally desorb and vaporize the collected particles during each 14-minute desorption cycle, with the filter temperature varying from 20 to 150 °C at a rate of 10 °C min$^{-1}$. The desorbed vapor analytes are sampled into a 150 mbar ion-molecule reactor and chemically ionized by iodide (I$^-$) ions generated by passing a gas stream containing CH$_3$I through a $^{210}$polonium radioactive source before

entering an LTOF mass analyzer for separation. The organic analytes are detected predominantly in the form of iodide adducts [M+I]$^-$ (> 95% relative abundance). The volume-weighted geometric mean diameters were determined using an SMPS (size range 9 - 834 nm, Leibniz Institute for Tropospheric Research, Germany). The SMPS(s) used for the measurements of chemical standards and $\alpha$-pinene SOA were calibrated using size standards of polystyrene latex beads.

## 2.4 Analyte particle surface coating

A 104 cm long Pyrex flow tube of 7.4 cm inner diameter with a total volume of approximately 5 L was used for analyte particle surface coating experiments (Figure S4) (Molteni et al., 2018). A 1000 ppm NH$_4$NO$_3$ solution in pure water was nebulized at 1.4 L min$^{-1}$ and dried before size-selection by the AAC. The resulting NH$_4$NO$_3$ particles passed through the charcoal denuders before entering concentrically into the flow tube with a laminar zero air sheath flow of 10 L min$^{-1}$ at 20 °C and 60% RH. Measurements were performed downstream of the flow tube. Particle composition and size were measured by the EESI-TOF

and SMPS (16 - 615 nm), respectively. Two different core sizes (155.8 and 226.4 nm) of NH$_4$NO$_3$ particles were used for coating experiments. 4.7 ± 0.4 ppm $\alpha$-pinene, as measured by a quadrupole proton-transfer-reaction mass spectrometer (Q-PTR), was injected into the reactor from a glass vial with a zero air carrier flow (1 L min$^{-1}$). To generate ozone, 20 - 200 mL min$^{-1}$ zero air (60% RH at 20 °C) was irradiated by an amalgam lamp (185 and 254 nm; WISAG GmbH, Switzerland). Ozone was mixed with $\alpha$-pinene to produce ozonolysis products which condensed onto, i.e. "coated" the NH$_4$NO$_3$ particles inside the

flow tube. Note that depending on the conditions this coating may either result in a core-shell structure or in formation of a homogeneous single phase, though the exact morphology does not affect the conclusion regarding surface extraction, as discussed below. The coating period in the flow tube was approximately 26 ± 0.5 s. The coating thickness was controlled by varying the ozone concentration in the presence of excess $\alpha$-pinene, which was measured by a Thermo 49A ozone analyzer (Thermo Fisher Scientific, US) to be 20 - 310 ppb. This ozone concentration range was optimized before injecting the NH$_4$NO$_3$

particles to ensure that no nucleation occurred which would have resulted in particles consisting only of SOA. At the beginning



of each ozone concentration step, the EESI-TOF sampled the gas and aerosol mixture through a bypass channel without denuder to ensure that all oxidation product signals reached steady state (< 20 min). Afterwards, routine sampling alternated between filtered background (5 min) and particle-phase measurements (10 min). This coating experiment was carefully designed to achieve high condensational growth rates of about 0.8 nm s$^{-1}$ with negligible nucleation.

## 3 Results and discussion

### 3.1 Influence of analyte particle size on EESI-TOF detection

Figure 1a shows a typical measurement of the EESI-TOF and SMPS for size-selected sucrose particles. Two sheath flow rates (5 and 15 with L min$^{-1}$) at 1.4 L min$^{-1}$ of particle flow were used to generate two different full width at half maximum size distribution for demonstrating the precision adjustment of AAC in size-selection. Due to the factor of 3 increase of the AAC sheath flow, the geometric standard deviation $\sigma_g$ of the size-selected sucrose particle distribution decreased from 1.4 to 1.2. A comparison of the signals in the red windows in Figure 1b shows that the sucrose signal did not increase commensurately with the volume concentration measured by the SMPS (regardless of $\sigma_g$), as the volumetric geometric mean diameter of the particles increased. To quantify this effect, we first define the size-dependent sensitivity $S(D_P)$ as

$$S(D_P) = \frac{I(D_P)}{M(D_P)},$$ (1)

where $I(D_P)$ is the intensity of the analyte normalized by the most abundant electrospray ion (NaINa$^+$) to account for ES fluctuation (< 5%) or the sum of fitted organic ions (Figure 3a). $M(D_p)$ is the mass concentration of the analyte measured by the SMPS or/and the AMS-LTOF as a function of the volume-weighted geometric mean mobility diameter $D_P$. To show the relative change of the sensitivity as a function of $D_P$, this sensitivity is normalized by the sensitivity at 100 nm electrical mobility diameter, defined as normalized sensitivity, $S_{100\ nm}$

$$S_{100\ nm} = \frac{S(D_P)}{S(D_P=100\ nm)},$$ (2)

where $S(D_P = 100\ nm)$ was interpolated with respect to the 3 parameters-fittings using the model function $S(D_P, P_1, P_2, P_3) = P_1 \cdot D_p\verb|^|(P_2) + P_3$. The normalization by the sensitivity at 100 nm is chosen to accommodate and compare all datasets in this study.

We investigated the normalized sensitivities of the EESI-TOF for levoglucosan, sucrose and NH$_4$NO$_3$ (tracers of biomass and anthropogenic activities in the ambient atmosphere) using different EESI ionization sources and ES operating parameters which resulted in different ES parent droplet diameters as detailed in Tables S1, S2 and S3. Figure 2 shows the normalized sensitivity of the size-selected analyte particles at 100 nm, (Eq. 2) as a function of the volumetric geometric mean diameter of the analyte particles generated using both pure component and mixed solutions detected under different ES conditions (see also Figure S6, Tables S1-3). The $S_{100\ nm}$ for different types of analyte particles decreases by up to 3 orders of magnitude as the volumetric geometric mean diameter increases from 30 to 300 nm, and some of them start to reach a plateau at larger sizes for experiments using EESI source A. The size-dependent sensitivity is observed for both single compounds and compound mixtures (Figure S6), indicating that the size-dependent sensitivity of the analyte particles is independent of their mixing state of different chemicals. We compared the $S_{100nm}$ to the calculated Brownian coagulation coefficient using three different ES parent droplet sizes (0.5, 1.5 and 5 μm) as described in SI Eq. 3-6. These sizes represent the estimated size ranges of ES parent droplets according to our ES operating parameters using scaling laws as described in Table S2 and Figure S7. The estimated Brownian coagulation coefficients are normalized to that for 100 nm monodisperse particles, analogous to the definition for $S_{100nm}$. We note that most measured normalized sensitivities correlate well with the normalized Brownian coagulation coefficient as shown in Figure 2. This implies that the longer the period for coagulation, the faster the saturation of the coagulated mass from smaller particle sizes that have higher Brownian coagulation coefficients. As a result, this higher period



for coagulation reduces the size dependent sensitivity variability after normalization to larger particle size that has lower Brownian coagulation coefficient. Furthermore, Figure 2 suggests that $S_{100nm}$ plateaus when the size-selected particles are reaching the vicinity of the estimated ES parent droplet size, as anticipated by the calculated Brownian coagulation coefficient normalized at 100 nm. The high deviation of size-dependent sensitivity (~50 %) after $D_p > 100$ nm is likely due to the differences in the size of ES droplets used in individual calibration runs that make up the composite data shown in Figure 2

(see Figure S6 an Tables S1-3), which affects the normalized Brownian coagulation coefficients as shown on the left-axis of Figure 2.

It is intuitive that the total coagulated mass is also dependent on the coagulation duration between the analyte particles and the ES droplets during electrospray ionization. Longer coagulation durations would allow for higher fractions of analyte particles to be extracted, i.e. the coagulation of smaller analyte particles would saturate, while the coagulation of larger analyte particles

would continue, which would result in shallower size-dependent sensitivities, i.e. smaller magnitude of size-dependent sensitivity. We examine this hypothesis by using an EESI source A which provides a factor of 2 longer residence time in the electrospray ionization region. As shown in Figure 2, the sensitivity size dependence resulting from EESI source B (red markers), which has twice the residence time as EESI source A, is significantly shallower than the one from EESI source A (blue markers), consistent with our hypothesis. Overall, Figure 2 suggests that the size-dependent sensitivity is limited by

Brownian coagulation, which varies with the ES droplet size and hence with the ES operating parameters, as well as the coagulation duration. Such size dependence would suggest that the ionization of analyte particles in the EESI proceed through the coagulation (partial coalescence) between analyte particles and ES droplets as reported by previous studies (Wang et al., 2012; Kumbhani et al., 2018; Pagonis et al., 2020).

Konermann et al. reported that the electrospray droplet evaporation can be affected by the size and the polarity of analyte

molecules (Konermann et al., 2013), while Meier et al. suggested that the extraction efficiency of EESI can depend on the volatility of analyte molecules (Meier et al., 2011). We investigated the EESI size dependence sensitivity for a complex mixture of internally mixed α-pinene oxidation products formed in the CLOUD chamber, to evaluate whether such dependence varies with the compounds' volatility, e.g. if volatile species preferentially evaporate from smaller analyte particles before their subsequent ionization. We generated unimodal size distributions of secondary organic aerosol (SOA) with volumetric

geometric mean diameters ranging from 17 to 137 nm (Figures S8 and S9). Figure 3a shows the normalized sensitivity of the sum of the detected organic ions measured by EESI-TOF after high-resolution peak fitting, $S_{100nm}$, as a function of the measured particle size. $S_{100nm}$ decreases from a value of 6 at $D_p = 17$ nm to ~1 at $D_p = 110$ nm. The relative change in normalized sensitivity is similar to the results obtained for individual chemical standards presented in Figure 2 for EESI source A. To examine whether there is a composition dependence on the EESI extraction, we compared the signals for the $C_{10}H_{16}O_{3-8}$

compounds measured by the EESI-TOF and the FIGAERO-CI-ToF-MS as shown in Figure 3b (see also Figure S10) from SOA formation at different temperatures in the CLOUD chamber. The signal of FIGAERO-CI-ToF-MS was integrated over the period of particle desorption. The sample collection efficiency of the filter used by FIGAERO is expected to be higher than 99.997 % for particles above 10 nm (Hilfiker-Lopez et al., 2014). The linear behaviours from different measured species between the EESI-TOF and the FIGAERO-CI-ToF-MS show that the relative abundances of the sampled aerosol chemical

composition are similar and comparable for both instruments with negligible re-volatilization of particles at two different sampling points. Thermal decomposition may affect the absolute quantification of particle-phase compounds by FIGAERO-CI-ToF-MS (Stark et al., 2017). To the best of our knowledge, no size dependence has been reported in the literature for this thermal artefact, which should be cancelled off after sensitivity normalization comparison in relative scale for each species. The sensitivity size dependence appears to be similar for $C_{10}H_{16}O_{3-8}$ compounds with estimated saturation vapor concentrations

ranging from $10^{-10}$ to $10^4$ µg m$^{-3}$. Both results from size-selected chemical mixtures and chemical resolution comparison between EESI-TOF and FIGAERO-CI-ToF-MS using a complex SOA mixture indicate that the EESI sensitivity size dependence is a function of the Brownian coagulation coefficient rather than molecular size, polarity, or volatility.


### 3.2 Influence of analyte particle coating thickness on EESI sensitivity

Limited surface extraction, approximately 2-4 nm in depth, of the analyte particles has been reported for some ESI source designs (Kumbhani et al., 2018; Wingen and Finlayson-Pitts 2019). If such an effect were present in the EESI-TOF design used in the current study, it could also appear as a size-dependent sensitivity. This would mean that a smaller fraction of the analyte volume is extracted as the analyte particle diameter increases, and that the EESI sensitivity scales with the analyte particle surface area rather than the volume. To determine the potential contribution by surface extraction to the observed sensitivity size dependence, we investigated the extraction efficiency of $NH_4NO_3$ particles, 156 and 226 nm in diameter without coating, which were coated with α-pinene oxidation products using source A. Source A was chosen because it has the greatest extent of size-dependent sensitivity in comparison to source B. This size range was chosen as the size-dependent sensitivity decreases by less than 15 % from 155 nm to 250 nm for single and mixed component analyte particles (Figure 2). The coating thickness on the $NH_4NO_3$ particles ranged between 12 and 26 nm, with a coated organic mass concentration up to 31 µg m$^{-3}$ (Figure S11). If extraction is limited by the analyte particle surface, the EESI signal for $NH_4NO_3$, i.e. $[NaNO_3+Na]^+$, should decrease with increasing coating thickness, similar to the size-dependent sensitivity (Figure 2) that is exhibited by the source A. In Figure 4, we show the signals of $NH_4NO_3$ and selected organic molecules with low volatility as a function of the coating volume (normalized to their respective minimum coating volume separately for each of the $NH_4NO_3$ particle core sizes). We find that the coating signal from $C_{10}H_{16}O_{6-8}$ is proportional to the coating volume, whereas the $NH_4NO_3$ particles signal remains constant with increasing coating thickness for both core sizes (see also Figure S12). This proportionality also demonstrates that the condensable species as coating substance is not limited by the mean oxidation states of oxidation products because there is no decrease of the $C_{10}H_{16}O_6$ for an increase of $C_{10}H_{16}O_8$. Our results suggest that there is no surface extraction limitation for analyte particles up to at least 250 nm in diameter for the EESI inlet designs used in the current study. Here, the coated analyte particles are probed using only EESI whereas the other study demonstrated surface extraction of the analyte particles using EESI and ESI, where the coating thickness was not measured and the dissolution period was several orders of magnitude shorter in their EESI than in ESI (Kumbhani et al., 2018). These differences might cause the discrepancies in comparison to our results.

### 4 Conclusion

We explored the EESI sensitivity response for size-selected analyte particles using individual chemical standards and chemical mixtures with two different EESI source designs. We show that the EESI sensitivity decreases as the size of the analyte particles increases. The sensitivity size dependence correlates with the Brownian coagulation coefficient, where the magnitude of the size-dependent sensitivity can be accounted for by the effective coagulated mass according to the coagulation duration between the ES parent droplets and the analyte particles. This suggests that the analyte particles undergo coalescence with the ES charged droplets as suggested in previous studies (Law et al., 2010; Wang et al., 2012), but the efficiency of the coalescence is limited by the coagulation rate attributed by the different analyte particle and ES charged droplet sizes. From a comparison with online FIGAERO-CI-ToF-MS with measurements we conclude that the EESI sensitivity size dependence is not affected by the volatility of the molecules for internally mixed secondary organic aerosols. Coating experiments show that the volume of analyte particles are fully extracted up to a size of 250 nm for our EESI configuration but the total extracted mass is limited by the size-dependent Brownian coagulation coefficients (i.e. not all analyte particles of different size can coalesce with all of the electrospray droplets) instead of limited surface extraction reported by the previous work (Kumbhani et al., 2018). Future work should investigate the EESI response to coarse mode particles (with $D_p > 1$ µm), where extraction may be limited to the analyte particles' surface. EESI users should be cognizant of the size-dependent sensitivity during their result interpretation. Such size dependence is especially relevant when studying aerosol formation and growth, and external mixtures of analyte



particles with largely different sizes. However, such effect is not expected to substantially influence the detection of ambient aerosols dominated by well-mixed accumulation mode particles.


*Data availability.* Data presented in this study can be obtained at the Zenodo online repository hosted by CERN (https://doi.org/ 10.5281/zenodo.4437079, Lee et al., 2021). Raw data can be obtained by contacting the corresponding authors.

*Author contributions.* C.P.L. and M.S. designed the experiment. M.S., C.P.L., D.W., H.L, M.W., F.A., V. H., B. L., performed
the experiments. C.P.L., M.S., M.W., F.A. analyzed the data. C.P.L., I.E., M.S., D.W., H. L., J. D., J.S. and U.B. interpreted the compiled results. C.P.L. prepared the manuscript. All authors contributed to the discussion and revision of the manuscript.

*Competing interests.* The authors declare that they have no conflict of interest.

*Acknowledgements.* We thank our technician Pascal Andre Schneider for technical support throughout our experiments. Special thanks to the CLOUD collaboration and CERN facilities for providing us the possibilities and resources to realize the investigation of the EESI extraction efficiency using complex analyte particles and comparison to other well-known aerosol mass spectrometry techniques.

*Financial support.* This work was financially supported by the Swiss National Science Foundation (20020_172602, BSSGI0_155846), the European Union's Horizon 2020 research and innovation program under the Marie Skłodowska-Curie grant agreement (No. 701647, No. 764991) and US National Science Foundation (AGS1801574, AGS1531284).

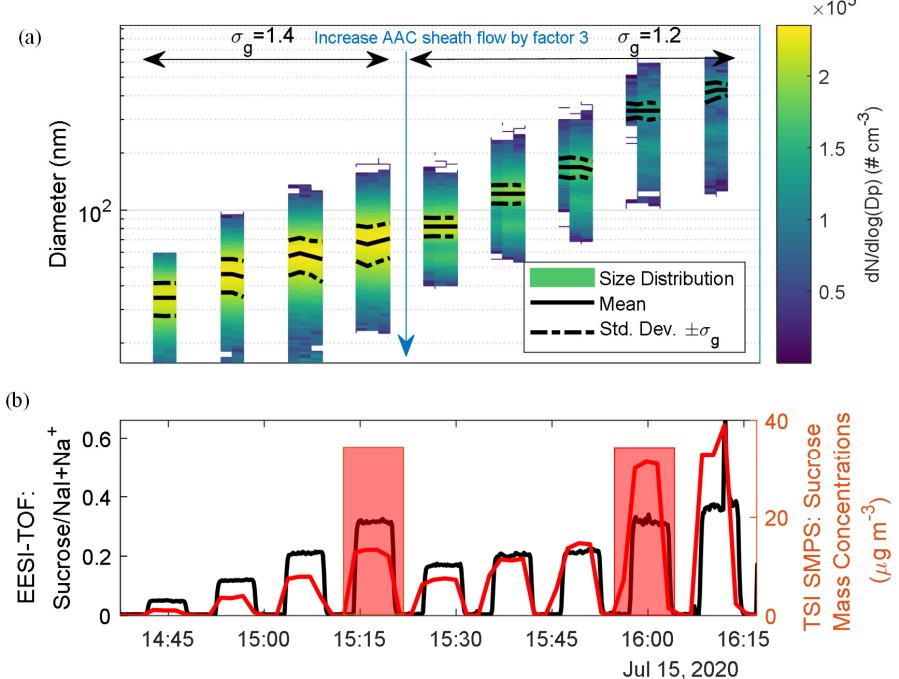

**Figure 1. (a) Measured number-weighted size distribution of sucrose particles after size selection using the AAC at two different**
**settings of the AAC sheath flow. The black solid line and the dotted lines denote geometric mean and standard deviation of the**
**number-weighted size distributions, respectively. $\sigma_g$ is the geometric standard deviation of the number-weighted size distribution.**
**The full width at half maximum (FWHM) of the size-selected particle distribution of sucrose is lower at the higher sheath flow of**
**the AAC. Data points of particle counts lower than 1 cm⁻³ were removed. Please see Figure S5 for the size-selection performance of**
**the AAC. (b) A representative EESI-TOF measurement showing the time series of sucrose normalized to the NaI+Na⁺ signal (ES**
**ion) and the corresponding integrated particle mass concentration measured by the TSI SMPS for size-selected sucrose particles**
**(using the integrated volume concentration and a density of 1.59 g cm⁻³). Red windows indicate periods with the same EESI signal**
**of sucrose but different size and mass concentration.**

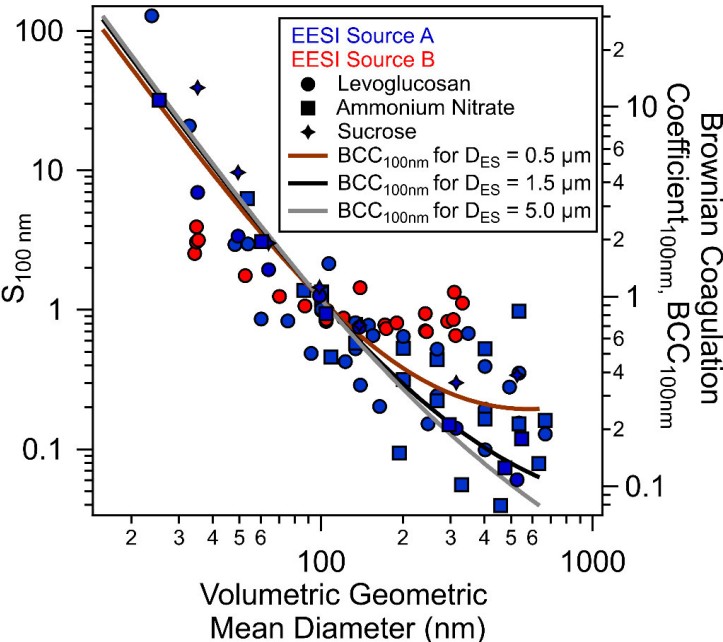

**Figure 2. Normalized sensitivity of EESI-TOF measurements at 100 nm as a function of volume-weighted geometric mean diameter. Red and blue markers indicate EESI source A and B which were developed for Orbitrap and TOF mass analyzers, respectively (Lopez-Hilfiker et al., 2019; Lee et al., 2020). Different marker types (●), (■), (✦) denote levoglucosan, NH₄NO₃ and sucrose, respectively. The Brownian coagulation coefficients are calculated using the range of ES parent droplet sizes according to our ES operating parameters (Figures S6 and S7). Note: Some of the data points may overlap at the similar volumetric geometric mean diameter due to repetitions of the same experiment settings.**



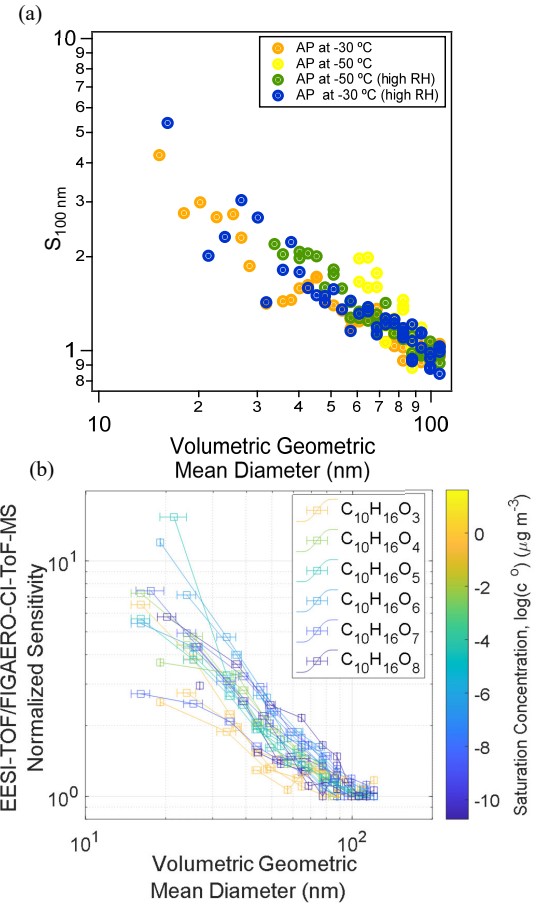

**Figure 3. (a)** Normalized sensitivity of EESI-TOF sum of high-resolution fitted organic ions as a function of the volume-weighted geometric mean diameter for each particle growth event in the α-pinene (AP) system at -30 and -50 ℃. **(b)** Normalized sensitivity of EESI-TOF intensity divided by the FIGAERO-CI-ToF-MS intensity for $C_{10}H_{16}O_{3-8}$ molecules in the particle phase as a function of the volume-weighted geometric mean diameter. Different marker types □, ✳, ○, ✚ indicate different SOA formation runs at -30 ℃, -50 ℃, -50 ℃ (high RH) and -30 ℃ (high RH), respectively. The saturation concentration was estimated as $log_{10}(C^0) = (n_C^0 - n_C)b_c - n_O b_O - 2 \cdot b_{CO}(n_C n_O)/(n_C + n_O)$ using the number of carbon and oxygen ($n_C, n_O$), interaction terms of carbon-carbon, oxygen-oxygen and carbon-oxygen ($b_c, b_O, b_{CO}$) and carbon number of alkane with a saturation vapor pressure of 1 μg m⁻³ ($n_C^0 = 25$) at 300 K (Donahue et al., 2011). See Figures S8, S9 and S10 for more information.




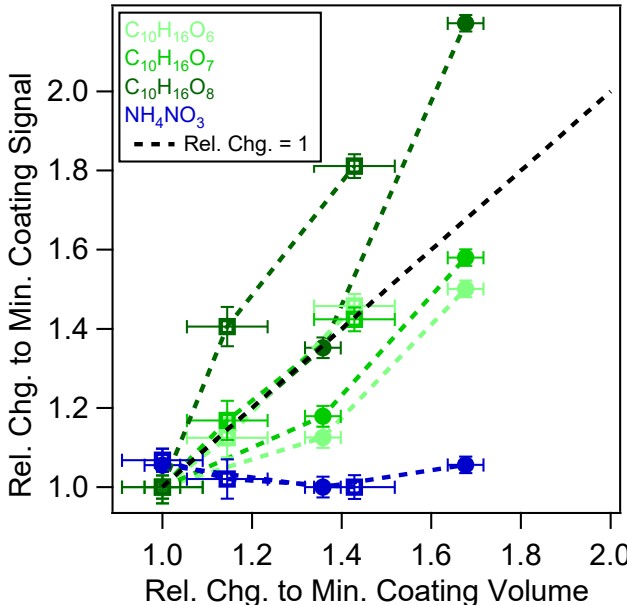


**Figure 4. Relative changes of α-pinene ozonolysis products coated on NH₄NO₃ particles at 156 (●) and 226 (■) nm core sizes. The coating volume (*x*-axis) measured by an SMPS is normalized by the smallest coating volume, and the coating signals (*y*-axis) of C₁₀H₁₆O₆₋₈ molecules as measured by the EESI-TOF are normalized by the signal at the smallest coating volume. The largest coating thicknesses are 19.8 and 26.8 nm for 156 and 226 nm core sizes of the NH₄NO₃ particles. The black-dotted line denotes a relative**

**change of the coating signal and volume equal to 1.**





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
