# Peer review of "Effects of Aerosol Size and Coating Thickness on the Molecular Detection using Extractive Electrospray Ionization"

_Atmospheric Measurement Techniques, 2021_

## Author Comment (AC1)

We thank the reviewers for their comments. The original comments are in normal font type. Our point-by-point responses are shown below in **bold** *and italics*. Manuscript changes are shown in blue and italics.

**Reviewer 1** (*https://doi.org/10.5194/amt-2021-22-RC1*)

Lee et al. present a characterization study of extractive electrospray ionization (EESI) of particles which is a topic of great interest given increased use of EESI to examine particles and their components. This EESI study of coated and uncoated particles of varied particle diameters is highly valuable given the range of results from existing EESI studies in the literature. The variety of methods of particle generation and compositions lends insight into the mechanism of extraction over a range of conditions that are greatly informative to the EESI community. The experimental techniques are robust and well-described and the paper is well written. The authors have clearly taken care in their experimental techniques, one example being the use of a size selection method that does not suffer from multiply charged particles.

Data are presented to show that Brownian coagulation between the analyte particles and the electrospray (ES) droplets can explain the reported increase in sensitivity for small particles because of increased time for coagulation for those smaller diameters. Brownian coagulation coefficients are shown to play a dominant role in controlling the sensitivity of EESI for SOA particles formed from pinene ozonolysis. A variety of particle compositions are shown to be fully extracted but with size dependence.

Main comments and questions are listed below:

1. The introduction mentions that the analyte particles contain <50% water. The experimental (section 2.3, line 117) describes the use of a silica gel dryer to dry particles, and section 2.4 (line 150) describes their humidification to 60% RH. Can it be clarified how the 50% water content (by mass?) was determined?

***Prior to injection into the flow tube, seed particles were dried using a silica gel dryer, where the RH is typically below 5%. The relative humidity of the sheath gas for the flow tube was maintained at 60%. The growth factor (ratio of wet to dry aerosol size, GF) of (dried) ammonium nitrate below 60% RH is ~1.2. In addition, SOA is slightly hygroscopic with GF typically below 1.2. Assuming that uptake of water vapour by analyte aerosol approaches equilibrium inside the flow tube, the water aerosol content may account for up to ~43%, which is consistent with our initial definition. However, as mentioned by Reviewer 2 Point 6, the sentence about water content definition is not necessary and can be confusing. Also, the droplets that are discharged from ES are now called ES droplets, particles that are injected into our EESI are called particles, and the analyte-laden droplets are the particles after reaction with the ES droplets. Thus, we removed the line 57-58 below***

***"Due to the lower water content of our analyte droplet in all our experiments (< 50%) than a typically known water content of droplet (> 90 %), we refer to our analyte droplets as analyte particles here hereafter."***

***and changed to***

"*For clarity, we refer to our neutral analyte droplets as "particles" prior to their interaction with ES droplets and as "analyte-laden droplets" afterwards.*"

Continuing the idea of particulate water content, this seems to be a large amount of water such that the particles are likely to have much greater internal mobility than dry, solid particles (even though many

particles cannot be completely dried).  The presence of particulate water could have a significant effect on their timeframe for solubility or coalescence within the ES droplets.  Since results are contrasted with a comparison paper, Kumbhani et al. 2018, which reports a seemingly large effect of the presence of water in their particles during EESI analysis, it is important to clarify this.  It is not clear if all the particles generated in Lee et al. have this much water, but if so this could be a further difference between results here and in the comparison papers that may be worth noting.

**Kumbhani et al. (2018) reports higher EESI sensitivity towards wet particles compared to dry particles, which is suggested to be the result of water mobilizing the surface layer. If EESI extraction is limited to the surface layer, increasing the aerosol water content may improve extraction. This hypothesis was derived from semi-quantitative comparisons of results obtained using online EESI vs. offline ESI, which differ in terms of their dissolution/extraction timescales (milliseconds during EESI vs. and minutes-to-hours during ESI) or (water) concentrations. In this study, we have shown for our EESI setup that the extraction is not limited to the surface layer. In addition, we have shown previously that the response factors do not vary significantly with respect to RH for our setup (Lopez-Hilfiker et al. 2019). We have added a new discussion of our findings in relation to Kumbhani et al. (2018).**

**In line 251-259: "*Aside from the size dependence, we did not observe any systematic RH influence on the EESI sensitivity from particle size selection (30 - 40 % RH) and SOA formation in the CLOUD chamber (20 - 60% RH) experiments. This is consistent with the findings by Lopez-Hilfiker et al. (2019), where RH does not systematically affect EESI sensitivity, but instead shows molecule-dependent effects where within an internally mixed particle ensemble the sensitivity of certain molecules may increase with RH while others decrease. The enhancement in EESI sensitivity for wet aerosol over dry aerosol was reported in a previous study (Kumbhani et al. 2018). If EESI extraction is limited to the surface of the analyte aerosol, the aerosol water content may mobilize surface species to facilitate dissolution (Kumbhani et al. 2018). However, the lack of RH dependence for our EESI setup indicates that such surface extraction limitation is absent in our study.*"**

**We have also added information regarding the RH conditions for our experiments as reference for future studies.**

**In line 129-133: *The high concentration of the chemical solutions ensured that sufficiently high analyte concentrations (> 3 μg m$^{-3}$) remain after size-selection by the AAC using the highest possible sheath flow (15 L min$^{-1}$ at an aerodynamic diameter $D_{ae}$ > 150 nm) to produce highly monodisperse particle distributions (Tavakoli and Olfert 2014) at 30 - 40 % RH. A HEPA filter was used for the background measurements after each particle size-selection.***

**In line 134-136: *In the second configuration (Figure S3), we investigated the size-dependent sensitivity of the EESI using biogenic SOA produced from α-pinene oxidation in the Cosmic Leaving OUtdoor Droplets (CLOUD) chamber at CERN, Switzerland ( Kirkby et al., 2011, Dias et al., 2017), at -50 to -30 °C and 20 - 60 % RH (Simon et al., 2020).***

2. Is it expected that the inorganic salt, NH$_4$NO$_3$, would be detected as [NaNO$_3$+Na]$^+$? It seems strange that no ammonium ions or ammoniated adducts are detected. If there is ion exchange from NH$_4$NO$_3$ particles with the NaI that is added to the electrospray solution, this is consistent with full extraction of analyte particles by the electrospray droplets.

**Na$^+$ is more electrophilic than NH$_4^+$ in the ion exchange and therefore bounds more strongly to the nitrate anion. In addition, NaI is present in great excess in the electrospray solution which promotes Na$^+$ adduct formation and suppresses other ionization pathways (e.g. [M+NH$_4$]$^+$ or [M+H]$^+$).**

But related to the question above, if the NH$_4$NO$_3$ was initially dried to lower than 50% water content, do the authors believe this ion exchange would still occur?

*Yes, our tests of using dried ammonium nitrate (without applying any humidification) indicate that $NH_4NO_3$ is detected predominately (> 99 %) as $[NaNO_3+Na]^+$ ions.*

3. I'm afraid I did not follow at first what the Brownian coagulation was referring to, although it is an insightful calculation. May I suggest that the authors use the same or similar wording used in the conclusion earlier on in the manuscript? Namely, include the description from line 289-290 "the coagulation duration between the ES parent droplets and the analyte particles" somewhere near line 196 where the Brownian coagulation coefficient was first described to clarify what interaction is being examined here.

*We refer to the Brownian coagulation as the possible mechanism that ES droplets and particles undergo during electrospray ionization. For a given Brownian coagulation coefficient, the total coagulated mass will vary with the ES droplet number concentration, analyte particle number concentration, and the time available for coagulation (i.e., residence time). After normalizing the Brownian coagulation coefficient and EESI sensitivity at 100nm as $BCC_{100nm}$ and $S_{100nm}$ in Figure 2, there is a systematic relationship between $S_{100nm}$ and $BCC_{100nm}$ for sizes below 200 nm and the slight discrepancies for sizes larger than 200 nm are clarified. We revised the discussion (line 200-216) as follows.*

*In line 200-216: If we assume that the detected ions from the size-selected particles in EESI are generated after coagulation and extraction between the particles and ES droplets, the normalized sensitivity $S_{100\ nm}$ is interpreted as the total coagulated mass. The determination of the total coagulated mass requires Brownian coagulation coefficients (BCC, computed from the particle and ES droplet sizes), number concentrations and residence time. However, the actual ES droplet size distribution could not be measured using other physical processes because these additional processes could alter the ES droplets properties and affect the electrospray ionization. Therefore, we could only calculate the BCC for different size-selected monodisperse particles assuming ES parent droplet sizes of 0.5, 1.5 or 5 µm. These three chosen ES parent droplet sizes represent the likely range of the actual ES droplet sizes, which is theoretically estimated from our ES operating parameters as summarized in Table S2, based on SI Eq. 3-6 and Figure S7. The calculated BCC values were normalized to the BCC for 100 nm monodisperse particles, denoted as $BCC_{100\ nm}$, as shown in Figure S6a, analogous to the normalization for $S_{100\ nm}$. Most normalized sensitivities (i.e. normalized total coagulated masses) correlate well with the $BCC_{100\ nm}$, as shown in Figure 2. Smaller particle sizes that have higher BCC are collected more efficiently and thus contribute a higher percentage of their total mass for extraction. Furthermore, the plateaus of $S_{100\ nm}$ at larger particles sizes could be explained by the suggested behavior of $BCC_{100\ nm}$ when the size of the particle is similar to the actual ES droplet size or partly to the estimated ES parent droplet size in our study. The high deviation of size-dependent sensitivity (~50 %) for $D_p > 200$ nm is likely due to the variation of the actual ES droplet size distribution in different calibration runs, which can deviate from the estimated ES parent droplet size. Knowledge of the actual ES droplet size distribution is needed to further explain the variabilities but are beyond the scope of the current study.*

4. Line 221 – The sentence states that EESI source A provides a factor of 2 longer residence time but then the next sentence (line 223) says source B has twice the residence time as source A. Can you clarify which one has the longer residence time?

*There is a typing error in line 220-222. We changed the phrase of "EESI source A" to "EESI source B": "We examined this hypothesis by using an EESI source B that provides a factor of 2 longer residence time in the electrospray ionization region."*

5. Figure S4 mentions the use of ammonium sulfate particles being coated with pinene oxidation products. Should this be ammonium nitrate?

*There is a typing error in the caption of Figure S4. We apologize for this. We changed the word "ammonium sulfate particles" to "ammonium nitrate particles": "Inorganic particles (NH₄NO₃) are injected into the flow tube which act as condensation sink for the low-volatility oxidation products."*

6. Figure S6, line 83, has "BCC", not sure if this is a typo?

*Yes, it is a typo; we have removed the additional "BCC".*

Also in Figure S6, the last sentence of the caption says that the mass concentrations for levoglucosan and $NH_4NO_3$ were measured by an LTOF-MS. Perhaps it would not change the trend of $S_{100\ nm}$ values in Fig. S6, but are there ionization efficiencies that affect the mass concentrations of levoglucosan and $NH_4NO_3$? There is limited description of how AMS signals were characterized in the main text and supplementary information.

*The ionization efficiencies are different for levoglucosan and $NH_3NO_3$ for the LTOF-AMS. But $S_{100nm}$ is calculated by normalizing the sensitivity of individual chemicals observed at different particle sizes to their respective observed sensitivity at 100nm, therefore, differences in ionization efficiencies between analytes should not affect the trend of $S_{100\ nm}$.*

**7.** It is quite nice to experimentally change the electrospray parent droplet size. The parent droplet sizes (0.7 – 5.66 um) seem larger than in the papers the authors compare with but I'm not sure this is always true. For example, Wang et al. (2012) points out that the ESI droplets are usually smaller than sample droplets and that this size is important in examining the mechanism. Could this be another difference between your studies and comparison papers (the conclusion does not refer back to this difference in drawing on comparisons).

*One of the differences between Wang et. al. (2012) and the current work is the size of the analyte droplets/particles. Wang et. al. (2012) used a dual spray setup, resulting in much larger analyte droplets ($D_P \gg 1\ \mu m$, 5-20 $\mu m$ ) compared to the study here ($D_p$ = 0.02 - 0.8 $\mu m$). While we expect the ES droplets to be similar or larger than the analyte particles in our study, the particles were shown to be overall larger than the ES droplets in Wang et. al. (2012).*

---

## Author Comment (AC2)

We thank the reviewers for their comments. The original comments are in normal font type. Our point-by-point responses are shown below in **bold** *and italics*. Manuscript changes are shown in blue and *italics*.
* * *
**Reviewer 2** (*https://doi.org/10.5194/amt-2021-22-RC2*)
* * *
This paper presents a series of carefully planned and executed experiments to explore the sensitivity of the EESI source as a function of particle size and composition, as well as EESI operating conditions. It represents a useful contribution to our understanding of the EESI, especially as it is becoming more commonly used in the measurement of ambient aerosol particles. However, some of the conclusions are not supported by the data and this needs to be corrected before publication.

Specific comments:

Discussion of the trend in normalized sensitivity on pages 5 and 6:

1. Line 197: Why do you state that the sensitivity reaches a plateau only for EESI Source A? The one set of data for EESI Source B shows a clear plateau.

***As requested, we added EESI source B in line 195-197:*** *"The $S_{100\,nm}$ for different types of analyte particles decreases by up to 3 orders of magnitude as the volumetric geometric mean diameter increases from 30 to 300 nm, and some of them start to reach a plateau at larger sizes for experiments using EESI source A and B."*

2. Line 217: What are you referring to as a longer period for coagulation? Do you mean the difference in length of interaction region in Source A and in Source B? If so, please state this more clearly. Source A has a 1 mm interaction region and Source B has 0.5 mm, so Source A has the longer period for coagulation. By your argument, Source A should have less of a size dependent sensitivity. The data in Figure 2 shows exactly the opposite. The blue points (Source A) have, taken as a whole, a steeper slope than the red points (Source B). Please draw a conclusion that it is supported by the data.

***There is a typing error in the figure and we apologize for this. We corrected the typing error of the interaction region distance denoted in Figure S1 where EESI source A should have 0.5 mm and EESI source B should have 1 mm, instead of the other way round. At the same time, we also corrected the typing mistake in line 220-222 as mentioned by Reviewer 1 at Point 4. After corrections of these 2 typing mistakes, the conclusion is supported by the data.***

3. Lines 210-216: None of these conclusions about plateaus or deviations are supported by the data. First, rephrase the sentence on lines 206 to 208 to make it clear that it is the calculation that suggests the sensitivity plateaus when the particles are close in size to the ES droplets.

***We clarified our interpretation of the coagulation coefficient in Line 210-216. In combination with the correction of the typing error described above (in 2), the conclusions are supported by the data:***

*"Smaller particle sizes that have higher BCC are collected more efficiently and thus contribute a higher percentage of their total mass for extraction. Furthermore, the plateaus of $S_{100\,nm}$ at larger particles sizes could be explained by the suggested behavior of $BCC_{100\,nm}$ when the size of the particle is similar to the actual ES droplet size or partly to the estimated ES parent droplet size in our study. The high deviation of size-dependent sensitivity (~50 %) for $D_p > 200$ nm is likely due to the variation of the actual ES droplet size distribution in different calibration runs, which can deviate from the estimated ES parent droplet size. Knowledge of the actual ES droplet size distribution is needed to further explain the variabilities but are beyond the scope of the current study."*

The data does not suggest this at all. In Figure S6, the data for the largest ES droplets plateaus at the smallest diameter, exactly the opposite of what the calculations suggest.

*The reviewer meant that one normalized sensitivity for the estimated ES parent droplet size of 5.66 µm in Figure S6c plateaus at the smallest diameter and it could be the opposite of what the Brownian coagulation coefficient calculation suggests. Please note that the ES droplet sizes meant by the Reviewer is an estimated ES parent droplet size using scaling laws (without accounting for the micrometer precision of the ESI capillary position) which we used as a theoretical reference to account for the possible variability range of our ES parameter settings. As this is an estimated ES parent droplet, it might not directly represent the actual ES droplet size that undergoes coagulation with the size-selected particle. For example, if there will be a slight change (± 5 % from 0.5 mm distance, i.e. ±25 µm) of the ESI capillary position, the actual ES droplet size could be smaller or larger than the estimated ES parent droplet size (5.66 µm) in the case of more or less time for evaporation, respectively.*

*To vary the actual ES droplet size in our experiments, we changed the ES capillary diameter (by a factor of 2), the ES capillary flow (a factor of 10) and ES voltages (20 %) as shown in Table S2. Such range of ES parameter settings should result in different actual ES droplet sizes that result in different normalized sensitivity behaviors. Despite all of these different normalized sensitivities, our whole experimental data demonstrates good correlation of $BCC_{100nm}$ and $S_{100nm}$ for sizes smaller than 200 nm and the normalized sensitivity plateaus when the particle sizes could be similar to the actual ES droplet size, as suggested by $BCC_{100\ nm}$. As a result, we clarified the possible discrepancies in line 212-215: "The high deviation of size-dependent sensitivity (~50 %) for $D_p$ > 200 nm is likely due to the variation of the actual ES droplet size distribution in different calibration runs, which can deviate from the estimated ES parent droplet size. Knowledge of the actual ES droplet size distribution is needed to further explain the variabilities but are beyond the scope of the current study."*

In addition, the sensitivity changes for two components of a single particle, e.g., levo and NO3 or sucrose and NO3, are very different on the sensitivity vs size graphs. For example, NO3 plateaus at ~ 250 nm and turns back up while levo from the same particles does not plateau at all. Sucrose plateaus at ~ 300 nm while NO3 from the same particles does not plateau at all.

*In order to generate enough particles for size-selection, the nebulization solution concentration was increased for size-selected standard calibrations conducted at larger sizes. It is possible that this led to small changes in the relative particle composition in the two-component mixtures. In addition, the ionization process (e.g. during droplet evaporation) and ion chemistry during the extraction process of EESI may also vary for different chemical species, especially so when comparing sugar/alcohol molecules (levoglucosan and sucrose) with an inorganic salt ($NH_4NO_3$). These differences on top of the prevailing size-dependent sensitivity merit systematic exploration in future studies (a step further), such as different chemical composition ratio at the same total coagulated mass, ES droplet and particle size distributions. This concern is admittedly under-constrained in the current study, as reflected in our manuscript changes above and below.*

I don't think you can draw any conclusions about particle size/droplet size relationships from the data. For the claim about the high deviation in the data above 100 nm, there is no discernible pattern as a function of ES droplet size in Figure S6. Therefore, it does not make sense to attribute the scatter in the data to ES droplet size. Or maybe you are saying that you have no idea what the droplet size is in any experiment. If that is the case, then state that.

*We did not measure (ultrasonically, optically or electrically) the actual electrospray droplets size, as doing so may affect the properties of electrospray droplets and the electrospray ionization. Thus, we agree with the reviewer that we do not have full control on the actual ES droplet size. Our approach was to perform theoretical calculations over a range of ES parent droplet sizes that is plausible according to a range of our ES operating parameters. As detailed in our response above, the theoretical calculation and experimental observation generally agree regarding the EESI sensitivity*

*dependence on the particle size. Therefore, we are confident that the actual size of ES droplet is an important parameter to explain the sensitivity behavior of plateau and scatter for larger particles. We have added a clarification regarding the ES droplet sizes as described above.*

4. Line 221: This statement that Source B has twice the residence time of Source A is not consistent with the schematic in Figure S1 and directly contradicts the preceding sentence.

*The typing error in line 220-222 was pointed out Reviewer 1 and we changed the phrase of "EESI source A" to "EESI source B": "We examined this hypothesis by using an EESI source B which provides a factor of 2 longer residence time in the electrospray ionization region."*

Residence time is not the explanation for the shallower sensitivity dependence of Source B. In addition, this (incorrect) residence time argument was already made in the previous paragraph and there is no need to repeat it here. Please remove this discussion.

*We corrected the errors in source labeling (A vs. B) in the main text and the supplementary information, and the implication of residence time should support the observed sensitivity dependence. A longer residence time for coagulation in EESI source B enables a higher extraction fraction of the particle and thus exhibits a less steep sensitivity size dependence.*

5. Lines 25-26: This sentence needs to be updated once you have revised the discussion sections. You do not demonstrate that the sensitivity dependence varies with ES droplet size or with residence time.

*This conclusion is supported by our analysis as explained in our responses to Point 1-4 above.*

Minor comments:

6. Lines 57-58: This sentence about water content is confusing – are the ES droplets really >90% water if you are using 50:50 H2O:ACN for the solution? You could just say you are going to call the analyte droplets particles to avoid confusion with the ES droplets. No need to invoke water content.

*We removed the sentence about water content definition as it is not very necessary. Now, the droplets that are discharged from ES are called as ES droplets, particles that are injected into our EESI are called particles, and the analyte-laden droplets are the particles after reaction with ES droplets. Thus, we removed the line 57-58*

*"Due to the lower water content of our analyte droplet in all our experiments (< 50%) than a typically known water content of droplet (> 90 %), we refer to our analyte droplets as analyte particles here hereafter."*

*and changed to*

*"For clarity, we refer to our neutral analyte droplets as "particles" prior to their interaction with ES droplets and as "analyte-laden droplets" afterwards."*

7. Line 65: What do you mean by "fragmentation of the analyte"? Isn't the point of EESI that it does not fragment the analyte so that you get molecular information.

*We think that this comment is referring to Line 64-66: "Using other techniques such as Dual- phase Doppler anemometer, Wang et al. suggested that the extraction happens via fragmentation of the analyte and ES droplets (Wang et al., 2012)", which describes the fragmentation of analyte droplets following droplet-droplet collision. Note that Wang et al., 2012 employed a dual-spray setup, where analytes are introduced as droplets with sizes similar to or larger than that of the ES droplets. We added a clarification to the main text in line 65-67: "...Wang et al. suggested that the extraction happens via fragmentation of the analyte droplets and ES droplets as the result of droplet-droplet collisions (Wang et al., 2012)."*

8. Line 132: The figures should be in the same order in the SI as you call them out in the text. Here you have S5 before S3 or S4.

*We removed sentence in line 132-133 because it was already mentioned in the caption of Figure 1. The SI figures are now in the correct order as referred in the text.*

9. Line 160: What do you mean by depending on conditions? What conditions and how do you know what morphology you have? You also say that the morphology will not affect your conclusions, "as discussed below" but you do not discuss morphology at all in Section 3.2. Please add a sentence or two about morphology to the discussion in Section 3.2.

*We mean that our main conclusions are not dependent on whether we have core-shell vs. single-phase particle morphologies. Assuming that extraction is limited to the particle surface, then either the core-shell or homogenous single-phase morphologies should result in the reduction of the $NH_4NO_3$ core signal as the organic coating thickness increases, but in different ways, whether due to the limited extraction "depth" or decreases in the $NH_4NO_3$ mass as a fraction of the coated particles, respectively. Because we did not observe reduction in the $NH_4NO_3$ signal during coating experiments, surface extraction limitations do not appear to be present, and thus our main conclusion should hold for either morphology. We added a clarification to this as follows:*

*Line 271-275: "If extraction were limited by the particle surface, the EESI signal for $NH_4NO_3$, i.e. $[NaNO_3+Na]^+$, should decrease similar to the size-dependent sensitivity (Figure 2) that is exhibited by the source A. If the coated particles were of core-shell morphology, then the extraction of the $NH_4NO_3$ core would be limited by the thickness of the organic coating and the ES extraction depth. If the coated particles were of homogeneous inorganic-organic mixture, then the detected $NH_4NO_3$ signal would decrease in proportion to the decreasing $NH_4NO_3$ mass fraction."*

10. Line 181: The figures should be in the same order as they are called out in the text. Here you have Figure 3a before Figure 2.

*We added "(Figure 2)" in line 180 before "(Figure 3a)" to keep the order.*

11. Line 284: What do you mean by dissolution period?

*We rephrased the line 282-285 that consists of dissolution period for clarity.*

12. Figure 2 caption: In the text, Source A and B are the TOF and Orbitrap, respectively. Please correct.

***We added the word "initially" and changed the order of TOF and Orbitrap in the Figure 2 caption:*** *"Blue and* yellow *markers indicate EESI source A and B which were* initially *developed for* TOF *and* Orbitrap *mass analyzers, respectively."*

13. Figure S1: Is the only difference between Source A and Source B the length of the gap between the ESI capillary and the transfer tube? Since you don't do any experiments with the Orbitrap, why show the schematic of it? I would move the inset for Source B to part A of the figure and delete part b of the figure.

***The reviewer is correct, the key difference between EESI Source A and B is distance between ESI capillary and the ion transfer capillary. Even though we did not use the Orbitrap in this study with source B, we decided to keep the schematic of the two sources separated to contextualize their design differences.***

14. Figure S2: In the schematic, you show a denuder and a HEPA filter, but you do not mention the use of the HEPA filter in the description of the experiments. You also don't mention bypassing the denuder as is shown in the schematic. Maybe you could simplify the red part of the schematic to match the text. Is the red arrow next to the HEPA filter in Figures S2 and S4 going in the wrong direction? Finally, please label the EESI.

***We added HEPA filter in the description of the experiments in line 132-133***: *"A HEPA filter was used for the background measurements after each analyte particles size-selection."*

***For simplification, we removed the channel that bypasses the denuder, corrected, and labelled the EESI in the Figure S2. However, Figure S4 remained the same because the bypass channel was required to observe the steady-state of the gas and particle composition for each step of coating as mentioned in the Section 2.4.***

15. Figure S6: Use the same symbols in 6d as in 6c for the same particles. In the caption, delete the extra "BCC" after the second sentence. Move the sentence about what A and B denote after the sentence describing panels b-d. It would be much easier to compare the data in these four panels if you use the same range on the x and y-axes.

***We updated the Figure S6 for the same range of x- and y-axes and its caption as requested.***

16. Table S2: You have reversed the Source labels A and B. Please correct. Why do you have two separate rows for experiments with mixed particles? For example Levo7 and AN2 are the same experiment, so just use one row.

*We have corrected the Source labels in Table S2 and combined the mixed particles into one row as mixed nebulization solution shown below.*

| Index no. | EESI Source Designs | ES voltage (kV) | ES pressure (mbar) | ES capillary inner diameter (um) | ES solution | Nebulization solution | ES Flow, Q (nl min$^{-1}$) |
|---|---|---|---|---|---|---|---|
| Levo1 + AN1 | A | 2.6 | 200 | 50 | $H_2O$:ACN (50:50 v/v) | Mixed | 354 |
| Levo2 + AN2 | A | 2.9 | 800 | 100 | $H_2O$:ACN (50:50 v/v) | Mixed | 22655 |
| Levo3 | A | 2.95 | 200 - 400 | 75 | $H_2O$:ACN (50:50 v/v) | Single | 1792-3584 |
| Levo4 | A | 2.88 | 200 - 400 | 75 | $H_2O$ | Single | 1309-2617 |
| AN3 | A | 2.95 | 200 - 400 | 75 | $H_2O$:ACN (50:50 v/v) | Single | 1792-3584 |
| AN4 + Suc1 | A | 2.9 | 600 | 75 | $H_2O$:ACN (50:50 v/v) | Mixed | 5375 |
| AN5 | A | 2.9 | 600 | 75 | $H_2O$:ACN (50:50 v/v) | Single | 5375 |
| Levo5 | B | 2.8 | 120 - 250 | 75 | $H_2O$:ACN (50:50 v/v) | Single | 1075-2240 |
| Levo6 | B | 3 | 120 - 250 | 75 | $H_2O$:ACN (50:50 v/v) | Single | 1075-2240 |
| Levo7 | B | 3 | 120 - 250 | 75 | $H_2O$:ACN (50:50 v/v) | Single | 1075-2240 |
| Levo8 | B | 2.9 | 120 - 250 | 75 | $H_2O$:ACN (50:50 v/v) | Single | 1075-2240 |
| Levo9 | B | 2.9 | 120 - 250 | 75 | $H_2O$:ACN (50:50 v/v) | Single | 1075-2240 |

17. Figure S8: What is the point of this figure? How is Figure S8 related to S9? I think you could skip S8.

*Figure S8 was referred in line 233-235 and is related to Figure 3 and Figure S9. The purpose of Figure S8 is to show the overview the EESI-TOF measurements and particle size distribution properties for consecutive SOA formation events in a very well-controlled chamber (CLOUD chamber at CERN). The shaded regions in Figure S8 denote the stage of the SOA formation where the size-dependent sensitivity of EESI is observed. These shaded regions were shown separately for of each SOA formation event in Figure S9. Thus, Figure S8 is related to Figure S9.*

18. Figure S12: This figure is not referenced in the main text. I think you could skip it.

*Figure S12 is referenced in line 279.*

19. There are many, many typographical errors (missing words, random extra words, misspellings, etc.). The authors should proofread much more carefully before submitting an article.

*We checked the texts and corrected typographical errors.*